# MTBP and MYC: A Dynamic Duo in Proliferation, Cancer, and Aging

**DOI:** 10.3390/biology11060881

**Published:** 2022-06-08

**Authors:** Brian C. Grieb, Christine M. Eischen

**Affiliations:** 1Vanderbilt-Ingram Cancer Center, Division of Hematology/Oncology, Department of Medicine, Vanderbilt University Medical Center, Nashville, TN 37232, USA; brian.grieb@vumc.org; 2Department of Cell & Developmental Biology, Vanderbilt University School of Medicine, Nashville, TN 37232, USA; 3Department of Cancer Biology and the Sidney Kimmel Cancer Center, Thomas Jefferson University, Philadelphia, PA 19107, USA

**Keywords:** MTBP, MYC, proliferation, cancer, transcription, aging

## Abstract

**Simple Summary:**

Cancer is the second leading cause of death globally, accounting for approximately one in six deaths. Due to its critical role in cell growth/proliferation, one of the most well-known cancer-promoting proteins is MYC. Because cancer cells need MYC to grow, MYC is usually overactive in human cancers. However, despite decades of research, drugs targeting MYC continue to remain elusive. Notably, MYC requires other proteins to function properly. Our investigations into a protein named MTBP revealed that it associates with MYC and helps its pro-growth and cancer promoting function. Here, we discuss MTBP and review evidence showing MTBP is a critical partner for MYC, although the full extent of this partnership remains unresolved. We also discuss the role of MTBP in cancer, as MTBP levels are elevated in many human cancers and is often associated with reduced cancer patient survival. Additionally, we will discuss the inhibition of cancer cell growth and induction of cancer cell death by decreasing MTBP levels, indicating MTBP may represent a new cancer drug target and a way of indirectly targeting MYC.

**Abstract:**

The oncogenic transcription factor c-MYC (MYC) is highly conserved across species and is frequently overexpressed or dysregulated in human cancers. MYC regulates a wide range of critical cellular and oncogenic activities including proliferation, metabolism, metastasis, apoptosis, and differentiation by transcriptionally activating or repressing the expression of a large number of genes. This activity of MYC is not carried out in isolation, instead relying on its association with a myriad of protein cofactors. We determined that MDM Two Binding Protein (MTBP) indirectly binds MYC and is a novel MYC transcriptional cofactor. MTBP promotes MYC-mediated transcriptional activity, proliferation, and cellular transformation by binding in a protein complex with MYC at MYC-bound promoters. This discovery provided critical context for data linking MTBP to aging as well as a rapidly expanding body of evidence demonstrating MTBP is overexpressed in many human malignancies, is often linked to poor patient outcomes, and is necessary for cancer cell survival. As such, MTBP represents a novel and potentially broad reaching oncologic drug target, particularly when MYC is dysregulated. Here we have reviewed the discovery of MTBP and the initial controversy with its function as well as its associations with proliferation, MYC, DNA replication, aging, and human cancer.

## 1. Introduction

MTBP was discovered over two decades ago in a yeast-two hybrid screen as a protein that bound to the p53 tumor suppressor regulator MDM2 [1]. Although initial investigations into the function of MTBP were focused on MDM2 [1,2], more recent studies have elucidated important functions for MTBP, far beyond what its name would suggest. It has been established MTBP functions as an oncogene, promoting proliferation and cellular transformation [3,4]. This behavior, at least in part, has been attributed to a transcriptional complex formed by MTBP with Tip48, Tip49 (also known as Pontin/RUVBL2 and Reptin/RUVBL1, respectively) and the oncogene c-MYC (MYC) [4]. MTBP has also been associated with DNA replication [5] and metastasis [6,7]. Consistent with its oncogenic role, MTBP has been shown to be overexpressed in a wide range of human cancers, where it is associated with poor patient outcomes in the majority of malignant contexts [4,8,9,10]. Moreover, suppression of MTBP expression in malignant cells or expression of a mutant with apparent dominant-negative activity demonstrated MTBP-directed therapeutics could have broad applicability as cancer therapies and extending patient survival [3,4,8,11], similar to its binding partner MYC. Herein, we review these advances and raise lingering questions.

## 2. MTBP, MDM2, and Early Controversy

The earliest investigation on MTBP focused on MDM2, due to its identification as a novel MDM2 binding protein—the source of its name [1]. MDM2 is a negative regulator of the tumor suppressor p53, which inhibits cancer development most notably by inducing cell cycle arrest and apoptosis in response to cellular stress [12]. With this in mind, Boyd et al. reported MTBP, like p53, induced a G1 cell-cycle arrest that was reversed by MDM2 overexpression [1]. However, a follow-up report indicated MTBP stabilized MDM2 and promoted MDM2-mediated degradation of p53 [2]. These seemingly conflicting reports spurred early controversy and motivated characterization of Mtbp function using genetically engineered mouse models. Deletion of *Mdm2* is embryonic lethal, but it is rescued by *p53* deletion, demonstrating the lethality is due to unchecked p53 activity [13,14]. However, although *Mtbp* deletion was also embryonic lethal, it could not be rescued by co-deletion of *p53* [6]. This observation brought into question whether MTBP regulated MDM2 as first reported [1,2].

Experiments with genetically engineered mice focused on tumorigenesis were utilized to further probe the connection between Mtbp and Mdm2, in vivo. Decreased Mdm2 expression in *Mdm2* heterozygous or hypomorphic mice delays tumor onset due to increased p53-mediated apoptosis, resulting in higher rates of p53 mutation and/or deletion in tumors that eventually develop [15,16]. In contrast to Mdm2, an *Mtbp* haploinsufficiency did not significantly change the rate of tumor development in wild-type or *p53*^+/−^ mice, nor did it alter the incidence of *p53* mutations/deletions [6]. In a separate analysis, *Mtbp* heterozygosity, like *Mdm2* heterozygosity, delayed Myc-driven B cell lymphoma development in the Eμ-*myc* mouse model, which overexpresses Myc specifically in B cells. However, the delay was determined not to be caused by increased p53 activity, and Mtbp status did not alter the incidence of *p53* mutations/deletions detected in the lymphomas that arose [3].

In the tumors analyzed from the above studies as well as in studies with wild-type or *Mtbp*^+/−^ thymocytes, Mtbp expression was not shown to correlate with Mdm2 or p53 expression [3,6], as first reported in vitro [1,2]. Collectively, these data indicated Mtbp does not regulate Mdm2 in vivo, particularly during tumorigenesis. Therefore, although the delay in lymphoma development with *Mtbp* heterozygosity indicates Mtbp has a role in cancer [3], the field was left with a fundamental question: What is the function of MTBP?

## 3. MTBP Promotes Cellular Proliferation and Transformation

Early data indicated MTBP was linked to cellular proliferation. For example, normal murine fibroblasts and malignant human cells that were serum-starved or forced to express cell cycle-arresting tumor suppressors exhibited decreased Mtbp/MTBP expression, while those provided serum or expressing pro-proliferative MYC or E2F1 exhibited increased Mtbp/MTBP expression [3]. These observations led to experiments in multiple publications examining the effects of both increased and decreased Mtbp/MTBP expression on proliferation in untransformed cells. For example, siRNA-mediated knockdown of Mtbp reduced proliferative capacity of *p53*-null mouse embryonic fibroblasts [3]. In contrast, Mtbp/MTBP overexpression increased cellular proliferation in a variety of epithelial cells and fibroblasts from humans, mice, and rats [4]. This same diametric effect on proliferation with reduced and increased MTBP expression was independently verified in HeLa, colon cancer, glioblastoma, and lung cancer cells [5,10,17,18,19].

In some instances, alteration of MTBP expression did not affect cellular proliferation. For example, neither Mtbp overexpression nor siRNA knockdown of *Mtbp* affected the proliferative capacity of primary murine osteosarcoma cells [6]. Another, perhaps more revealing, example includes the observation that *Mtbp* heterozygosity did not affect proliferation of primary murine wild-type pre-B cells [3]. However, when more rapidly proliferating untransformed Eμ-*myc* pre-B cells that overexpress Myc were cultured under the same conditions, those heterozygous for *Mtbp* exhibited reduced proliferative capacity compared to wild-type Eμ-*myc* pre-B cells [3]. This suggests a mismatch in MTBP expression and proliferative rate is required for MTBP to affect proliferation. In other words, cells proliferating slowly may tolerate lower levels of MTBP, whereas low MTBP expression can be limiting to cells with higher rates of proliferation. Regardless, the preponderance of evidence from multiple groups demonstrates MTBP expression changes in response to growth signals to actively promote proliferation, not merely to support it. While the specific pathway(s) that activates MTBP for proliferation is unknown, MTBP has even been shown to be transcriptionally activated by MYC in multiple cellular contexts [3,19], further supporting its pro-proliferative function.

Beyond cell proliferation, overexpression of Mtbp in murine fibroblasts was sufficient to induce cellular transformation, as measured by soft agar colony formation and tumor development in mice [4]. The effects of Mtbp in this context are modest, especially compared to a powerful oncogene like Myc. Nevertheless, increased MTBP expression is also able to promote neurosphere formation [19]. Additionally, compared to wild-type littermates, *Mtbp* heterozygous mice had reduced spontaneous cancer development [11]. Taken together, the data indicate that MTBP has oncogenic activity.

## 4. MTBP as a MYC Transcriptional Cofactor

The mechanism through which MTBP promotes cellular proliferation and transformation was brought to light by our discovery that MTBP serves as a cofactor for MYC [4]. MYC is a highly conserved multifunctional basic helix–loop–helix leucine zipper (bHLH-LZ) oncogenic transcription factor that is estimated to regulate 10–15% of the genome, enacting a transcriptional program to control critical cellular processes, with proliferation and cellular transformation being chief among them [20,21,22,23,24,25,26,27]. MYC does not conduct this transcriptional orchestra in isolation, instead relying on a myriad of cofactors. These cofactors associate with MYC through highly conserved amino acid sequences known as MYC Boxes (MB) that are numbered I, II, IIIa, IIIb, and IV [27].

While their conservation alone suggests all the MB regions contribute to MYC function, arguably the most important is MBII [27,28]. Mutants of MYC lacking MBII are deficient in MYC transcriptional and downstream biologic activities, particularly MYC-induced cellular transformation, both in vitro and in vivo [29,30,31,32]. Cofactors that require MBII for binding to MYC are critically important for MYC function. Reduced association of cofactors that bind the MBII domain of MYC or catalytically inactive mutants of the cofactors are well known to reduce MYC transcriptional activity [33,34,35,36]. For example, independent of the MYC cofactor TRRAP, the ATPases Tip48 and Tip49 bind the MBII domain and regulate MYC [34,37,38,39,40,41]. A missense mutation in the Tip49 enzymatic domain functions as a dominant negative inhibitor of MYC, reducing its oncogenic activity [34]. As depicted in Figure 1, we characterized an association between MTBP and MYC, demonstrating it was mediated by the C-terminus of MTBP and the MBII domain of MYC [4]. However, this interaction between MTBP and MYC was indirect and was mediated by Tip48 and Tip49 [4].

Consistent with an N-terminal nuclear localization sequence and predominant nuclear distribution of MTBP [3,17], chromatin immunoprecipitation (ChIP) as well as ChIP-reChIP experiments revealed MTBP associated with MYC at its pro-proliferative target genes, including both MYC-activated and repressed genes [4]. These results are consistent with reports demonstrating Tip48 and Tip49 associate with MYC regulated promoters in multiple cell types and species [35,37,38,39,40]. In these first targeted ChIP experiments interrogating MTBP association with chromatin, MTBP associated at all MYC target genes tested, but not at specific regions of the genome unbound by MYC [4]. Whole genome assessment of dual occupancy for MYC and MTBP at promoters has not been reported, so the true extent of overlap between MTBP and MYC occupancy at a genome level remains unknown. However, CUT&RUN [42] has been reported for MTBP, and this showed over 30,000 unique binding sites residing mainly in promoters, enhancers, and super enhancers, as would also be expected of MYC [43]. For example, MTBP associating with chromatin was focused around H3K4me2, a mark of MYC activated gene targets [44,45]. Coincidentally, these areas are also known to function as points of DNA replication initiation [46,47,48,49,50,51,52,53,54,55], another function of MTBP discussed below. Further testing and evaluation are required to determine which chromatin sites of MTBP association can be explained by co-localization with MYC versus DNA replication origins or perhaps other as yet unknown factors.

Investigations into the function of MTBP as a MYC cofactor revealed MTBP facilitated MYC-mediated transcriptional activation of pro-proliferative genes. *Mtbp* heterozygosity or *MTBP* knockdown by siRNA limited the ability of MYC to activate highly conserved pro-proliferative MYC transcriptional target genes [3,4], whereas MTBP overexpression enhanced MYC-mediated transcription [4]. With this mechanistic backdrop and the interaction between MTBP and MYC at MYC-bound promoter regions, it is unsurprising that decreased MTBP expression limits MYC-driven proliferation, while MTBP overexpression enhances MYC-mediated proliferation [3,4]. In addition, a fragment of MTBP comprising the C-terminal third of the MTBP protein readily inhibited MYC-dependent transcription and downstream proliferation [4]. While the C-terminal region has been separately implicated in DNA replication ([56], discussed below), the C-terminal MTBP fragment that inhibits MYC retains the ability to associate with both the Tip48-Tip49-MYC complex and MYC-bound promoter regions [4]. Although MYC cofactors, such as Tip49, limit MYC activity if enzymatically inactive [33,34,35], no enzymatic activity has been attributed to MTBP. Therefore, the remaining N-terminal two-thirds of MTBP likely contains one or more protein-recruiting/binding domains to bring other proteins into the transcription complex to enable MYC to enact its pro-proliferative transcriptional program. Additional experimental investigation is needed to further elucidate the protein–protein interactions by MTBP and their role in MYC-mediated transcription.

While the exact biochemical mechanism by which MTBP enhances MYC transcriptional function remains unresolved, clues may be garnered by considering their shared association with Tip48 and Tip49. The cooperation between MTBP and MYC in transcription, proliferation, and cellular transformation through their association via Tip48 and Tip49 has been established. Moreover, both Tip48 and Tip49 proteins have been linked to processes and protein complexes that epigenetically modify DNA, allowing for transcriptional activation and repression [41]. For example, Tip48 and Tip49 are in complex with the histone acetyl transferase Tip60 and are necessary for its function, enabling modulation of MYC-transcriptional activity [35,57]. Moreover, Tip48 and Tip49 were implicated in histone variant switching, opening DNA at promoters and transcriptional start sites [58]. Future investigations are needed to determine whether MTBP has an influence on recruiting or regulating epigenetic modifiers that associate with Tip48-Tip49-MYC complexes that control chromatin opening and/or closing to aid MYC-induced transcriptional activation or repression, respectively. Furthermore, Tip48 and Tip49 have been described in other transcriptional complexes, such as E2F1 and β-catenin/TCF [41,59,60,61]. It is unknown whether MTBP has an undiscovered role in these oncogenic transcriptional complexes, or whether its activity is isolated to MYC.

The consequences of MTBP allowing MYC to effectively transcribe its target genes, inducing proliferation, should increase MYC-mediated cellular transformation when MTBP is overexpressed or dysregulated. In fact, MTBP was shown to enhance MYC-mediated cellular transformation (Figure 1). Specifically in mice, untransformed mouse fibroblasts stably expressing increased levels of both Myc and Mtbp formed larger tumors after subcutaneous injection than those overexpressing Myc or Mtbp alone [4]. In vitro colony assays also showed analogous results of cooperation between Mtbp and Myc in transformation [4]. Of note, a significant barrier to MYC-induced cellular transformation is apoptosis, which occurs with overexpressed or dysregulated MYC. In untransformed cells, overexpressed or dysregulated MYC induces apoptosis by indirectly suppressing the expression of anti-apoptotic proteins such as BCL2, BCLX, and BCLW and by activating pro-apoptotic proteins such as p53 [62,63,64,65,66,67,68,69,70]. MTBP overexpression has been shown to limit MYC-induced apoptosis [4], indicating MTBP may be enlisted by MYC to overcome this critical barrier to oncogenic transformation. Because MTBP promotes pro-proliferative MYC-dependent transcriptional activity and limits MYC-induced apoptosis in untransformed cells, these may provide cells with sufficient time to disable the effects of tumor suppressors that otherwise restrain MYC-mediated cellular transformation.

## 5. MTBP in DNA Replication

Related but distinct to its function in proliferation is the role of MTBP in DNA replication origin firing. MTBP has been reported to be critical for the final step in the initiation of DNA replication and has been characterized as the metazoan ortholog to yeast Sld7 [5]. As shown in the bottom of Figure 1, MTBP is reported to form a complex with Treslin, TopBP1, and cyclin-dependent kinase 8/19–cyclinC (Cdk8/19-cyclin C) that enables recruitment of Cdc45 to DNA, completing assembly of the activated replication helicase CMG (Cdc45-Mcm2-7-GINS) [71,72]. This process is enabled by the N-terminus of MTBP that associates with Treslin and the central domain that associates with Cdk8/19-cyclin C. The C-terminus, known in this context as the CTM, appears to function as a DNA binding domain [56]. In studies with *Xenopus* egg extracts, the kinase DDK facilitated the association between MTBP-Treslin with chromatin and with TopBP1 [73]. Moreover, six phosphorylation sites targeted by cyclin dependent kinases and Cdk8/19-cyclin C as well as 14 sites targeted by DNA damage checkpoint kinases allow MTBP to enhance or inhibit DNA replication, respectively [74]. Functionally, knockdown of *MTBP* is reported to prolong S-phase in HeLa, U2OS, and HCT116 cells as well as *Xenopus* oocytes [5,56]. This leads to subsequent inaccurate chromosome separation during G2/M. These results support an early report that *MTBP* siRNA caused chromosome missegregation, although this was attributed to decreased MTBP-mediated recruitment of Mad2 to chromosome kinetochores [17].

While the model of MTBP in DNA replication via a Treslin-Cdk8/19-CycC-TopBP1 complex has been established, Tip48, Tip49, and MYC also have connections to DNA replication and chromosome segregation. Specifically, in conjunction with the ATP-dependent chromatin remodeler INO80, the MTBP binding partners Tip48 and Tip49 stabilize stalled replication forks, and deletion of INO80 has been shown to delay S-phase progression [75,76,77]. Additionally, inhibition of Tip48 and Tip49 enzymatic ATPase activity in lung cancer also resulted in stalled replication and prolonged S-phase, sensitizing these cells to radiation [78]. Interestingly, a similar sensitization to radiation was observed in glioblastoma cells following knockdown of *MTBP* [19]. Moreover, Tip48 and Tip49 are involved in the proper assembly and organization of microtubules and thereby chromosome separation during mitosis [79].

With regards to MYC and DNA replication, MYC directly regulates DNA replication by associating with the pre-replication complex and facilitating DNA replication initiation [80,81,82]. Moreover, MYC has been shown to maintain its association with chromatin during mitosis [83], offering an alternate explanation as to why MTBP is detectable at chromatin in prometaphase [17]. Dysregulation of MYC results in an abnormal G2/M checkpoint and subsequent errors in chromosome segregation [84,85,86,87]. Furthermore, downmodulation of MTBP was shown to mimic a decrease in Mad2 expression [17], which has been shown to be a transcriptional activation target of MYC [87]. Thus, while the function of MTBP in the DNA replication machinery has been identified, its relationship with Tip48/Tip49 and MYC may additionally contribute to DNA replication.

## 6. MTBP Impacts Aging

MTBP has also been described as a modifier of the aging process [11], providing additional support for its role in cellular proliferation and MYC-mediated transcription (Figure 1). Mice heterozygous for *Myc* display increased longevity [88]. This is largely related to the role MYC has in increasing cellular metabolism and protein translation, well known causes of aging [89,90,91]. MYC also controls other cellular processes classically associated with aging, such as replicative stress, senescence, apoptosis, stem cell maintenance, genomic instability, and metabolism [28,92,93,94]. For example, MYC has been shown to increase reactive oxygen species (ROS) formation [95,96], and decrease tissue repair over time [97,98]. MYC has also been reported to regulate or be regulated by prominent proteins in the aging field, such as mTOR and SIRT1 [99,100].

Consistent with its role in limiting Myc activity, *Mtbp* heterozygous mice exhibited a 20% increase in lifespan compared to littermate matched wild-type control mice [11]. The longevity of mice with reduced Mtbp expression was partially attributed to a delay in spontaneous cancer development, fitting with the long-standing role of Myc as an oncogene and the ability of Mtbp to regulate Myc-driven cellular transformation [4]. Moreover, there is growing evidence that development of aging-related metabolic diseases (e.g., type 2 diabetes mellitus and hyperlipidemia) is linked to increased risk of cancer development [101], suggesting Mtbp may be modulating age-related tumorigenesis through modulating Myc-regulated metabolism. Consistent with this idea, the livers of long-lived *Mtbp*^+/−^ mice were more metabolically active at a molecular level. For example, livers in *Mtbp*^+/−^ mice expressed twice as much *Pcg1α* and *Pcg1β* RNA, markers of mitochondrial health [102,103,104,105,106]. These livers also had elevated expression of the well-known anti-aging gene *Sirt1* [107,108,109]. Interestingly, with respect to Myc, the livers of long-lived Mtbp heterozygous mice had higher levels of important Myc metabolic targets genes, such as *Ncl*, *Cad*, and *Odc* [11]. This same pattern was not observed in skeletal muscle or brown fat, suggesting that at physiologic levels, MTBP may not always be limiting to MYC transcriptional activity in all organs or cellular contexts, or at all times during physiologic development and/or aging. Although further investigation will be required to define the tissue-specific physiologic roles for MTBP, the observations that either low Myc or low Mtbp expression promotes longevity in mice provide additional support that MTBP functions with MYC.

## 7. MTBP Contributions to Cancer

Given the function of MTBP in proliferation, DNA replication, and cellular transformation as well as the connection between MTBP and MYC, it would be expected that MTBP has a critical role in cancer. Moreover, in cell culture, MTBP increases proliferation when growth factors are limiting [4], suggesting that sustained or dysregulated MTBP expression could serve as a mechanism for cells to survive and proliferate with limited external growth signals, echoing a hallmark of cancer [110]. Indeed, MTBP expression appears to be selected for during the process of cellular transformation. This was first observed in lymphoma where Mtbp/MTBP was discovered to be overexpressed in primary murine B cell lymphomas as well as human B cell lymphoma cell lines [3]. Moreover, publicly available data indicate *MTBP* mRNA is overexpressed in multiple human cancers including breast, cervical, colorectal, gastric, lung, prostate, and squamous cell carcinoma of the skin [4] (see Table 1). *MTBP* mRNA and protein were reported to be overexpressed in a panel of human colon cancer cell lines as well as 60 primary colon cancer samples with matched normal controls (*p* < 0.01) [10]. Early-stage lung adenocarcinoma samples (*n* = 119) exhibited high *MTBP* mRNA compared to normal lung tissue (*p* = 0.0069) [9]. MTBP protein was also observed to be elevated in a panel of human breast cancer cell lines [8].

While in some situations this increased expression of MTBP may be due to the high proliferative state in these cancers and/or MYC dysregulation, the *MTBP* gene undergoes amplification in multiple cancers. A comprehensive analysis of data from The Cancer Genome Atlas (TCGA) indicated *MTBP* is amplified in at least 20 different types of human cancer [4]. Notable examples include 32% of ovarian cancers, 18% of breast cancers, 19% of hepatocellular carcinomas (HCC), 7% of lung cancers, and 5% of colorectal cancers. Other research groups have independently reported *MTBP* amplifications in colorectal carcinoma and multiple myeloma [116,117], as well as multiple human cancer cell lines [118]. Notably, *MTBP* amplifications frequently co-occur with *MYC* amplifications, despite the two genes being separated by over 7.2 megabases on chromosome 8 [4], providing further support for the selection of MTBP overexpression when MYC is dysregulated.

Regardless of how MTBP is overexpressed, data indicate high MTBP expression correlates with poor patient outcomes in the majority of clinical contexts (Table 1). In breast cancer patients, both elevated *MTBP* mRNA expression (*n* = 842, *p* = 0.0337) and *MTBP* gene amplification (*n* = 913, *p* = 0.0195) independently predict poor patient survival [8]. Similarly, high MTBP protein expression was associated with decreased patient survival (*p* = 0.024) and with resistance to sorafenib therapy (*p* = 0.0025) in a panel of 120 and 52 HCC patients, respectively [112,114]. Furthermore, several studies have demonstrated the importance of MTBP in lung cancer. In an analysis of multiple datasets of stage I lung adenocarcinoma patients, MTBP expression was associated with decreased patient survival (*p* < 0.001, *p* = 0.001 and *p* = 0.037) [18]. To validate this finding, MTBP protein levels were assessed in a cohort of 99 patients with stage I lung adenocarcinoma, and MTBP overexpression was associated with shorter overall survival of patients (*p* = 0.041) [18]. In a separate analysis of MTBP protein expression by immunohistochemistry in 112 lung adenocarcinoma patient samples spanning stage I to stage IV disease, “hyper-expression” of MTBP was observed in 23.21% of samples [9]. The elevated expression was associated with stage IV disease (*p* = 0.008), the presence of malignant pleural effusions (*p* = 0.02), and decreased patient survival (*p* < 0.001).

In a broader analysis of publicly available datasets, high *MTBP* mRNA expression was associated with worse survival for patients with lung cancer (*p* = 0.01011 and 1.745 × 10^−5^) [9], consistent with the studies above. These analyses also demonstrated increased *MTBP* expression was associated with poor patient outcomes in glioblastomas (*p* = 7.344 × 10^−5^), gliomas (*p* = 4.4342 × 10^−12^ and 3.053 × 10^−5^), and renal cancer (*p* = 1.062 × 10^−5^). A subsequent study in glioblastoma confirmed increased *MTBP* expression was associated with higher grade gliomas and poor patient outcomes [19]. Here, elevated *MTBP* expression correlated with the Classical molecular subtype of glioblastoma, known to carry poorer outcomes than the Proneural molecular subtype [119].

Interestingly, even at the level of cancer patient survival, the observation that MTBP is a limiting factor for MYC oncogenic activity can be observed. MYC transcriptional activity is commonly linked to poor patient outcomes [120,121]. In breast cancer patients (*n* = 421) whose cancers have high levels of *MYC* mRNA expression, concurrent high expression of both *MYC* and *MTBP* mRNA conferred worse survival compared to those with high *MYC* and low *MTBP* expression (*p* = 0.0314) [4]. Similar trends were observed for colorectal (*n* = 215, *p* = 0.151) and lung adenocarcinomas (*n* = 174, *p* = 0.1331) [4]. Moreover, in breast cancer, *MTBP* mRNA levels were the highest in the aggressive and deadly triple negative breast cancer (TNBC) subtype [8,122,123], which has been reported to have a high MYC transcriptional signature [120]. Not only do these observations support the clinical significance of MTBP, but they also support its role in regulating MYC transcriptional activity in malignant cells.

While increased MYC expression is often linked to poor patient prognosis [124,125,126,127,128], there are examples where this does not appear to be the case [121,129]. In these instances, *MYC* mRNA and even protein expression may not reflect the ability of MYC to enact its transcriptional program, due to the complex multilayered regulation of MYC expression and activity [27]. Instead, evaluating the expression of over 350 MYC target genes to determine MYC transcriptional activity has been shown to more accurately predict cancer patient outcomes [120]. The interaction between MYC and MTBP in controlling patient survival suggests simply examining MYC expression in the context of MTBP expression could also serve as a surrogate measure of MYC transcriptional activity and more accurately predict patient outcomes than MYC expression alone. To our knowledge, such an analysis has not been reported, either with mRNA expression data or immunohistochemistry.

Nevertheless, there may be some clinical or tissue-specific scenarios where increased MTBP expression confers better patient outcomes. For example, MTBP levels evaluated by IHC in 129 patients with squamous cell carcinoma of the head and neck did not correlate with overall survival (*p* = 0.815), although low MTBP expression was associated with decreased survival in a narrowly defined subset of patients (*n* = 55, *p* = 0.004) that the authors had already characterized as having decreased survival independent of MTBP [115]. Moreover, reduced MTBP expression has been linked to increased invasion and migration of tumor cells in culture and metastasis in mouse models [6,7]. This has also been observed clinically. In a panel of 352 gastric cancer patients, high MTBP protein levels were associated with improved patient survival (multivariate analysis hazard ratio 0.633, 95% confidence interval [CI] 0.417–0.961, *p* = 0.032) with low levels of MTBP being associated with higher rates of both lymph node (*n* = 273, odds ratio [OR] 0.282, 95% CI 0.161–0.494, *p* < 0.001) and distant metastases (*n* = 48, OR 0.365, 95% CI 0.138–0.965, *p* = 0.042) [111]. Additionally, *MTBP* mRNA expression was reported to be lower in 20 HCC samples compared to matched adjacent normal liver tissue (*p* < 0.01) [113]. This same group performed IHC for MTBP in 102 patient HCC samples, which demonstrated low levels of MTBP were associated with capsular/vascular invasion (*p* < 0.001) and lymph node metastasis (*p* < 0.05); however, overall survival was not assessed [113].

The reports linking increased MTBP levels to improved outcomes are difficult to interpret among the preponderance of data showing MTBP expression is associated with poor patient survival. However, cancer cells can downregulate MYC and proliferation in favor of movement [130]. Both studies by Bi et al. in HCC and Wang et al. in gastric adenocarcinoma characterized MTBP as a metastasis suppressor [111,113]. These results conflict with observations from early-stage lung cancer, colon cancer, and a separate HCC study, where MTBP overexpression was associated with metastasis [10,18,112]. Moreover, Bi et al., utilized HCC samples collected exclusively from early-stage patients who underwent primary resection, whereas the other HCC studies discussed above examined more clinically advanced disease that required locoregional and/or systemic therapy beyond surgery. Based on these discrepancies, it is unclear if MTBP has tissue- or stage-specific behaviors. Moreover, it is not known if MTBP regulates oncogenic activities associated with MYC in a context-specific manner. Therefore, although the preponderance of current evidence indicates that MTBP expression is commonly selected for by multiple types of cancer and confers poor prognosis, cancer cell proliferation and metastasis have been shown to have a dichotomous interaction [130] that further complicates the contribution of MTBP to cancer. Thus, further investigation is needed to flesh out these different roles of MTBP in cancer proliferation and metastasis.

## 8. Targeting MTBP for Cancer Therapy

Although clearly important in malignant cells, the therapeutic potential of MTBP is still unknown (Figure 1). The current data suggest it may be an effective drug target. In TNBC cell lines, shRNA-mediated knockdown of *MTBP* significantly reduced proliferative capacity [8]. Notably, *MTBP* knockdown did not increase the proportion of cells in late S or G2/M phases of the cell cycle, as was reported in HeLa cells [5,17]. Instead, loss of MTBP induced apoptosis as measured by Annexin V positivity, cleavage of caspase 3, and sub-G1 DNA accumulation [8]. The growth of TNBC xenografts in nude mice were also significantly inhibited, even after tumors were allowed to establish [8]. This was the first in vivo study to demonstrate the potential of MTBP as a therapeutic target in cancer.

Investigations into the therapeutic potential of MTBP have extended beyond breast cancer. Knockdown of *MTBP* by shRNA or siRNA decreased proliferation of lung cancer cells [4,18] and reduced colony formation of gastric adenocarcinoma cells [131], respectively. Additionally, MTBP was identified in a CRISPR-Cas9 screen as one of nine essential genes for gastric adenocarcinoma survival [131]. The effects of reducing MTBP expression have also been examined in glioblastoma, where knockdown inhibited proliferation and neurosphere formation as well as induction of apoptosis [19]. Mice with intracranial glioblastoma xenografts expressing *MTBP* shRNA demonstrated significantly increased survival compared to mice with tumors expressing control shRNA. Interestingly, shRNA knockdown of *MTBP* sensitized glioblastoma xenografts to both radiation and the alkylating agent temozolomide, suggesting targeting MTBP could synergize with existing therapies.

Collectively, these data indicate that targeting MTBP may provide therapeutic benefit, particularly in TNBC, glioblastoma, and lung cancer. However, further testing is required to define the full spectrum of cancers that might benefit from therapies directed against MTBP. While our own efforts focused on TNBC, we also reported MTBP expression is elevated in estrogen receptor positive (ER+) breast cancers as well as those expressing the receptor tyrosine kinase human epidermal growth factor receptor 2 (HER2) [8]. Moreover, there are many other cancers that actively select for MTBP overexpression as well as cancers that rely on high MYC activity [27,132,133]. As such, MTBP may represent an as yet untapped broadly applicable therapeutic target in human cancer, or an avenue to augment the effectiveness of existing therapies.

Targeting MTBP effectively may rely on its C-terminus. As stated above, a C-terminal fragment of MTBP that mediates its association with MYC also appeared to function as a dominant negative inhibitor of MYC transcriptional and pro-proliferative activity [4]. This same fragment also inhibited the growth of TNBC cell lines [8]. A compound able to mimic this C-terminal MTBP peptide could serve as a small molecule inhibitor to target MYC oncogene addiction in many human cancers [132]. This would be highly significant because reducing MYC activity or expression has been shown to have a therapeutic benefit in several different forms of cancer [132,134,135,136,137,138,139] and even in cancers not driven by MYC itself [140,141]. Moreover, given the prominent role of the C-terminus of MTBP in regulating DNA replication, this approach to targeting MTBP may further inhibit the rapid rate of DNA replication in cancer cells, essential for cancer cell proliferation.

With regards to MYC, the C-terminal MTBP mutant could be inhibiting MYC activity by displacing endogenous MTBP from the MTBP-Tip48-Tip49-MYC complex, inhibiting proper formation/function of the complex at chromatin. Alternatively, the mutant could be sponging away and reducing the pool of another unidentified cofactor in the complex. Further investigation into the specific amino acids responsible for the binding of MTBP to Tip48/Tip49 as well as crystallization of this structure may allow for development of potential therapeutics to target MTBP in cancer, as has been accomplished with other MYC cofactors [142,143,144,145,146,147,148,149,150]. Alternatively, other reports demonstrate that decreasing the expression or interfering with the ATPase function of Tip48/Tip49 in HCC decreases tumor growth and viability [151,152]. Investigation of therapeutic strategies to disrupt the ATPase function of Tip48/Tip49 as well as the association between MYC and Tip48/Tip49 or the assembly of Tip48/Tip49 hexamers/dodecomers might also have value. However, it is unclear if these drug development efforts would also target MTBP’s role in DNA replication, further emphasizing the importance of evaluating the interplay of the known functional complexes containing MTBP.

The efficacy of these strategies for treating cancer cannot be accurately predicted, but a concern for each would be toxicity. Knockout of *Myc*, *Tip48*, *Tip49*, or *Mtbp* are each embryonic lethal [6,37,153,154]. MYC also regulates the growth and maintenance of many normal tissues throughout the body [155,156]. Fortunately, mouse models examining the effects of whole body *Myc* inhibition suggest these theoretical treatments may be tolerable [140], although care should be taken in extending these conclusions to human patients as well as to Tip48/Tip49, which have functions beyond MYC [41]. With regards to MTBP specifically, *Mtbp^+/−^* mice showed no overt defects or abnormalities and displayed increased longevity [11]. Additionally, an *Mtbp* haploinsufficiency was adequate to significantly limit Myc-driven B cell lymphoma development and proliferation [3]. Taken together, these data indicate that inhibition of MTBP to a level sufficient to provide therapeutic effects is likely tolerable.

## 9. Conclusions

Since its discovery over 20 years ago, great progress has been made in the characterization of MTBP. It has risen from an obscure protein thought erroneously to only regulate its namesake MDM2 to a prominent protein in cellular proliferation, transformation, and DNA replication. These functions have been linked primarily to novel chromatin-bound protein–protein complexes, Tip48, Tip49, and MYC as well as Treslin, TopBP1, and Cdk8/19-cyclin C. With regards to MYC, MTBP associates with MYC at MYC-regulated promoter regions, promotes oncogenic MYC-transcriptional activity, enhances MYC-driven cellular proliferation and transformation, confers worse prognosis for patients with elevated MYC expression, and is critical for the survival of malignant cells with high MYC transcriptional activity. However, two major questions remaining in the field are whether the MTBP-MYC and the MTBP-Treslin models can be harmoniously linked or if they are separate, and what is the precise biochemical mechanism by which MTBP regulates MYC-mediated transcription. It will also be important to identify any other protein complexes, if they exist, with which MTBP associates.

Beyond its molecular characterization, several groups have identified MTBP as an important protein in multiple human cancers. Although individual analyses suggest MTBP may have tissue- and/or context-specific behaviors in malignant cells, the preponderance of data indicate MTBP is a clinically significant factor that correlates with worse patient outcomes. Reducing MTBP expression or activity has also been shown to limit proliferation and/or induce apoptosis in several malignant contexts. Therefore, the data indicate that MTBP should be considered a potential target for therapeutic purposes. Furthermore, MTBP expression may be exploited as a prognostic marker for patient survival or as a predictive marker to inform therapy selection. Therefore, further investigation of MTBP in cancer and development of therapies targeting it are potentially beneficial and warranted.

## Figures and Tables

**Figure 1 biology-11-00881-f001:**
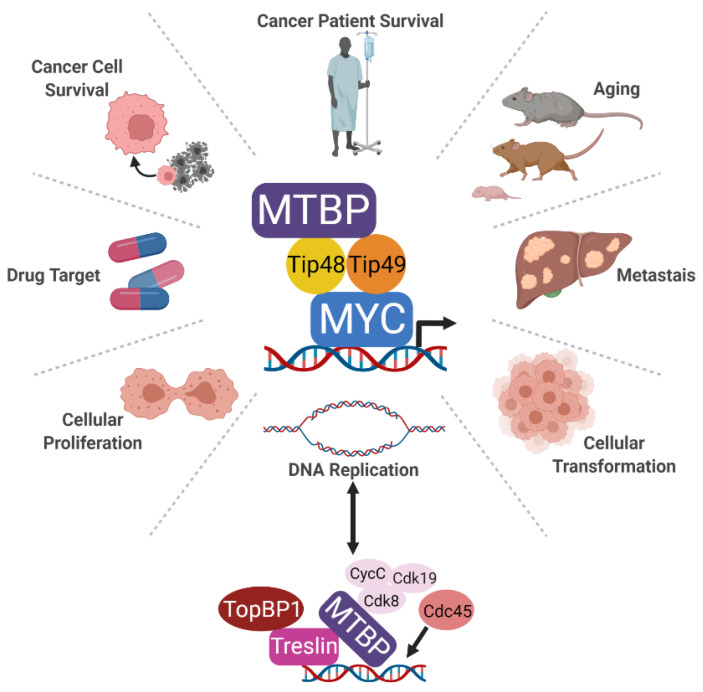
**Schematic of the identified MTPB complexes and the cellular processes for which MTBP function has been associated.** MTBP forms a complex with MYC at MYC-bound promoters via a direct interaction with Tip48 and Tip49. Research has linked MTBP to DNA replication, proliferation, cellular transformation, cancer cell survival, and metastasis as well as aging. While all of these functions can be linked back to MYC, MTBP additionally forms a complex with Treslin to promote the recruitment of Cdc45, enabling DNA replication origin firing. Similar to MYC, these collective oncogenic activities make MTBP an important potential cancer drug target.

**Table 1 biology-11-00881-t001:** Expression of MTBP mRNA or protein in human cancers and correlation with patient overall survival.

Cancer Type	MTBP Up	MTBP Down	High MTBP and Patient Survival	Patients (*n*) *	Publication
Breast	mRNA		Decreased	639	Grieb et al., 2014 [8]
Cervical	mRNA		ND	20	Grieb et al., 2014 [4]
Colorectal	mRNA		ND	273	Grieb et al., 2014 [4]
Colorectal	mRNA/Protein		ND	60/ **	Shayimu et al., 2021 [10]
Gastric Adenocarcinoma	mRNA		ND	91	Grieb et al., 2014 [4]
Gastric Adenocarcinoma		mRNA/ Protein	Increased	20/352	Wang et al., 2017 [111]
Glioblastoma	mRNA		ND	81	Grieb et al., 2014 [4]
Glioblastoma	mRNA		Decreased	337	Song et al., 2019 [19]
Glioma/Glioblastoma	mRNA		Decreased	1440	Mao et al., 2018 [9]
Hepatocellular Carcinoma	mRNA/Protein		Decreased	120 ***	Lu et al., 2015 [112]
Hepatocellular Carcinoma		mRNA/ Protein	ND ^#^	20/102	Bi et al., 2015 [113]
Hepatocellular Carcinoma	mRNA		Decreased	52 ^##^	Jiang et al., 2021 [114]
NSCLC Adenocarcinoma	mRNA/Protein		Decreased	739/112	Mao et al., 2018 [9]
NSCLC Adenocarcinoma	mRNA		ND	45	Grieb et al., 2014 [4]
NSCLC Adenocarcinoma (Stage 1)	mRNA/Protein		Decreased	119/99	Pan et al., 2018 [18]
NSCLC Squamous Cell Carcinoma	mRNA		ND	27	Grieb et al., 2014 [4]
Prostate Carcinoma	mRNA		ND	13	Grieb et al., 2014 [4]
Renal Cell Carcinoma	mRNA		Decreased	792	Mao et al., 2018 [9]
Squamous Cell (Head & Neck)	Protein		No Change	184	Vlatkovic et al., 2011 [115]
Squamous Cell Carcinoma (Skin)	mRNA		ND	11	Grieb et al., 2014 [4]

ND, Not Determined; * those studies that evaluated mRNA and protein, the number of patient samples for each is separated by a slash; ** number of samples evaluated for protein is unclear; *** number of samples evaluated for mRNA expression versus protein expression is not explicitly stated; ^#^ low MTBP expression was associated with lymph node metastases; ^##^ all patients treated with sorafenib.

## Data Availability

Not applicable.

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
