# Peer review of "MTBP and MYC: A Dynamic Duo in Proliferation, Cancer, and Aging"

_biology, 2022, doi:10.3390/biology11060881_

Round 1

Reviewer 1 Report

I read with great interest the manuscript, which falls within the aim of this Journal. In my honest opinion, the topic is interesting enough to attract the readers’ attention. Nevertheless, authors should clarify some points and improve the discussion, as suggested below.

Authors should consider the following recommendations:

  • Manuscript should be further revised in order to correct some typos and improve style.
  • Accumulating evidence suggests that obesity and metabolic diseases (such as type 2 diabetes mellitus, dyslipidaemia, and cardiovascular diseases) may play a key role in increasing the risk of cancers, modulating pivotal cross-talk pathways for cell proliferation and differentiation. I recommend to stress these important points, referring to: PMID: 27347932; PMID: 28115924.

Author Response

We thank the reviewer for his/her comments. We have addressed your comments point-by-point below.

Manuscript should be further revised in order to correct some typos and improve style.

Response:

We apologize for any typos. We have reviewed the manuscript and corrected the typos we identified and will work with the editorial staff of the journal to correct other typos we may have missed.

Accumulating evidence suggests that obesity and metabolic diseases (such as type 2 diabetes mellitus, dyslipidaemia, and cardiovascular diseases) may play a key role in increasing the risk of cancers, modulating pivotal cross-talk pathways for cell proliferation and differentiation. I recommend to stress these important points, referring to: PMID: 27347932; PMID: 28115924.

Response:

We have) highlighted the importance of metabolic disease and cancer development in section 6 (MTBP Impacts Aging) to emphasize this point and added a reference.

Reviewer 2 Report

This review article is well written and covers both the historical aspects and the frontiers of the functional/clinical interaction between two enigmatic cancer-related proteins, MTBP and MYC. The citation of the relevant articles is fair. Here are several suggestions that the authors feel free to consider to improve the impact of the discussion.

Simple summar (add ‘y’ to make it ‘summary’)

This reviewer is sure that the authors know how widely or specifically MTBP is involved in MYC biology. If so, please briefly mention it here. Obviously, MTBP has MYC-independent functions; conversely, MYC also has MTBP-independent functions. If so, the authors should mention this realistic aspect in this section. Otherwise, the words in the main title, “dynamic duo,” could be a little misleading or overspeculating as it sounds very attractive to this reviewer and probably many prospective readers. 

Abstract

  1. “a highly conserved protein”: This reviewer wonders where and how they are highly conserved. The authors may add “within the MYC family of transcription factors” or “among different animal species.”
  2. “….including proliferation, metabolism, metastasis, and differentiation”: You had better add “apoptosis” here.
  3. “…. indirect MYC binding partner”: “indirect” and “binding partner” conceptually sound a little conflicting with one another. “……is a dynamic component of the MYC protein complex when operating” might be more appropriate. Please consider.
  4. “….As such, MTBP represents a novel and potentially broad-reaching oncologic drug target…..”: Why? Just because MTBP is overexpressed in cancers where MYC is deregulated? If so, this description seems a little too broad to convey your excitement to the readers. Again, if you know a certain cancer case (or a mechanism) by which MYC needs MTBP (or vice versa), say so briefly here and add the following words at the end of this sentence. “, particularly where MYC is hard to inhibit.” Alternatively, “..., particularly where MYC is overexpressed.” If MTBP and MYC form a “dynamic duo,” the authors may include a possible future direction for anticancer drug development based on a new influential collaboration between MYC and MTBP in the Abstract.

  1. Introduction

  1. “……or expression of non-functional mutants demonstrated……”: “non-functional” means no activity. If possible, “dominant-negative mutants” or “gain-of-function mutants” seem more appropriate here.
  2. In Table 1. the authors argue that “the preponderance of data indicates it (= MTBP overexpression) is associated with poor patient outcomes.” According to Table 1, it seems true in some human cancers, such as breast cancer, glioblastoma, liver cancer, lung cancer, and renal cancer. Indeed, in each paper cited in this table, the number of patients was quite large, so the result seems to be statistically significant. To verify the result of the table, this reviewer quickly checked the updated Kaplan-Meier survival curve data of the patients of various human malignancies with high MTBP expression vs. low MTBP expression available online at “Human Protein Atlas.” It demonstrates that the tendency depends on the type of cancer, but it does not apply to all human cancers. This discrepancy is also true in the case of MYC. Considering that the possible “dynamic duo” between MTBP and MYC in human cancer progression is the main theme of this review article, it would be interesting to collect online data on both MTBP and MYC (including N-MYC and L-MYC) in various human cancers and discuss something more in-depth here (in addition to the current Table 1). If there is any correlation or collaboration between overexpressed MTBP and overexpressed MYC in poor prognosis, even if not in all human cancers, that would be more informative and pertinent to this potentially interesting review article.

  1. MTBP, MDM2, and early controversy

“…… Mtbp does not regulate Mdm2 in vivo, particularly during tumorigenesis……. What is the function of MTBP?.....”: Even if those data in vivo showed that Mtbp does not regulate Mdm2, those data do not theoretically exclude the possibility that Mdm2 regulates Mtbp function. Thus, to clarify the upstream and downstream functions, the last sentence should be written as “What is the effector function of MTBP?”

  1. MTBP as an MYC Transcriptional Cofactor (“a MYC” does not have to be “an MYC”?)

  1. Ref [27] contains insufficient information.
  2. “….. Cofactors binding MBII are critically important for MYC function….”: MBII is a highly hydrophobic short peptide region, so it is unlikely that the MBII region was adequate to isolate an “MBII-binding” protein. Thus, this sentence may be written as “Cofactors that require MBII for binding MYC are critically important for MYC function.” In the paper [35], Tip48 and Tip49 fail to bind to the MYC protein that lacks MBII.
  3. In this section 4, there is no description of TRRAP (Transformation/Transcription Domain Associated Protein), which was originally identified as an MYC-binding protein requiring MBII that promotes MYC transactivation and transformation. Is TRRAP entirely replaced by Tip48/Tip49 when mediating MYC-induced transcription and transformation in the presence of MTBP? If so, briefly mention it and cite the paper. Otherwise, the authors should discuss any overlapped function among these MYC-binding proteins (Tip48, Tip49, and TRRAP) requiring MBII in the presence and absence of MTBP.

  1. MTBP in DNA Replication

“…..Specifically, in conjunction with INO80, the MTBP-binding partners Tip48 and Tip49 stabilize stalled replication forks, and….”: This is the first and only place discussing INO80 in this review article, so the authors can briefly describe INO80. For example, “a DNA helicase-related ATPase subunit, INO80” instead of just “INO80.”

  1. MTBP contributions in cancer

  1. “….. Regardless of how MTBP is overexpressed, data indicate high MTBP expression correlates with poor patient outcomes in the majority of clinical contexts (Table 1)…..” : As discussed above (see 1-(b)), the Kaplan-Meier curve demonstrates that in some cancers, such as glioma (inconsistent with ref[19]) and head/neck cancer, reduced MTBP-expressing cancer is accompanied by much poorer survival than increased MTBP-expressing cancer. There is almost no survival difference between +/- high MTBP populations in colorectal cancer within the first six years (according to the online data via the “Human Protein Atlas”). Thus, the current sentences using “in the majority of clinical contexts” seem a little misleading. Re-phrase it.
  2. “The interaction between MYC and MTBP in controlling patient survival suggests simply examining MYC expression in the context of MTBP expression could also serve as a surrogate measure of MYC transcriptional activity and more accurately predict patient outcomes than MYC expression alone. To our knowledge, such an analysis has not been reported, either with mRNA expression data or immunohistochemistry.” : This reviewer is glad to read this paragraph as it indicates a more valuable prognostic value to detect both MYC and MTBP than either one alone. Please include this important message in the Abstract (see above: Abstract-(b)).

  1. Targeting MTBP for Cancer Therapy

  1. “….The current data suggest it may be an effective drug target. In TNBC cell lines, shRNA-mediated knockdown of MTBP….”: If TNBC appears here first, spell it out (i.e., triple-negative breast cancer (TNBC)).
  2. “…..Investigation of therapeutic strategies to disrupt the ATPase function of Tip48/Tip49 as well as the association between MYC and Tip48/Tip49, or the assembly of Tip48/Tip49 hexamers/dodecomers might also have value. However, it is unclear if these drug development efforts would also target MTBP’s role in DNA replication…..”: This is a fair statement, but it will probably need to be followed by additional sentences indicating what needs to be done next. For example, “…..However, it is unclear if these drug development efforts would also target MTBP’s role in DNA replication. Thus, it will be pertinent to clarify whether Tip48/Tip49 facilitates non-transcriptional DNA replication by MYC…..”
